evolution/palaeontology

Ediacaran, trace fossil, ecosystem engineering, Cambrian, extinction

**Author for correspondence:**
Alison T. Cribb
e-mail: cribb@usc.edu

# Increase in metazoan ecosystem engineering prior to the Ediacaran–Cambrian boundary in the Nama Group, Namibia

Alison T. Cribb[1,2], Charlotte G. Kenchington[3], Bryce Koester[1,4], Brandt M. Gibson[1], Thomas H. Boag[5], Rachel A. Racicot[1], Helke Mocke[6], Marc Laflamme[7] and Simon A. F. Darroch[1]

[1]Earth and Environmental Science, Vanderbilt University, Nashville, TN 37235-1805, USA
[2]Earth Sciences, University of Southern California, Los Angeles, CA 90089-0740, USA
[3]Earth Sciences, University of Cambridge, Cambridge CB2 3EQ, UK
[4]Department of Biodiversity, Drexel University, Philadelphia, PA, 19104, USA
[5]Geological Sciences, Stanford University, Stanford, CA 94304, USA
[6]Geological Survey of Namibia, Ministry of Mines and Energy, Windhoek, Namibia
[7]Chemical and Physical Sciences, University of Toronto Mississauga, Mississauga, Ontario, Canada L5L 1C6

ATC, 0000-0002-8604-6100; CGK, 0000-0002-8201-8786; BMG, 0000-0001-9620-0063

The disappearance of the soft-bodied Ediacara biota at the Ediacaran–Cambrian boundary potentially represents the earliest mass extinction of complex life, although the precise driver(s) of this extinction remain unresolved. The 'biotic replacement' model proposes that an evolutionary radiation of metazoan ecosystem engineers in the latest Ediacaran profoundly altered marine palaeoenvironments, resulting in the extinction of Ediacara biota and setting the stage for the subsequent Cambrian Explosion. However, metazoan ecosystem engineering across the Ediacaran–Cambrian transition has yet to be quantified. Here, we test this key tenet of the biotic replacement model by characterizing the intensity of metazoan bioturbation and ecosystem engineering in trace fossil assemblages throughout the latest Ediacaran Nama Group in southern Namibia. The results illustrate a dramatic increase in both bioturbation and ecosystem engineering intensity in the latest Ediacaran, prior to the Cambrian boundary. Moreover, our analyses demonstrate that the highest-impact ecosystem engineering behaviours were present well before the onset of

the Cambrian. These data provide the first support for a fundamental prediction of the biotic replacement model, and evidence for a direct link between the early evolution of ecosystem engineering and the extinction of the Ediacara biota.

# 1. Introduction

The terminal Neoproterozoic Ediacaran Period (635–539 Ma; [1,2]) represents a critical interval in Earth history, marking the first appearance of ecosystems dominated by complex eukaryotic, soft-bodied macroscopic organisms (colloquially referred to as the 'Ediacara biota'). The taxonomic affinities of the soft-bodied Ediacara biota have been much debated, although recent work suggests they represent a mixture of stem- and crown-group animals, as well as extinct clades with no modern representatives [3–7]. After approximately 30 Myr of ecological dominance, the overwhelming majority of soft-bodied Ediacaran groups (sometimes defined within clades—see [8,9]; although, see [10]) decline in the latest Ediacaran Nama interval and disappear entirely at the Ediacaran–Cambrian boundary, potentially representing the first mass extinction of complex life [6,11–15]. Establishing the drivers of this extinction is thus key to understanding the origins of the modern, animal-dominated biosphere.

Three major models have been proposed to explain the disappearance of the Ediacara biota: (i) a 'catastrophe' model, which proposes a global-scale environmental perturbation analogous to the 'Big 5' Phanerozoic mass extinctions (e.g. [11]), (ii) a 'biotic replacement' model, which proposes that the extinction was the result of intensifying ecosystem engineering from emerging Cambrian-type metazoan fauna [12], and (iii) a 'Cheshire Cat' model, which proposes that the disappearance of the Ediacara biota is instead due to a taphomoic bias from the loss of non-actualistic preservational environments related to the microbial matgrounds [12]. Whereas the 'Cheshire Cat' model has been refuted by work showing that Ediacaran-style matgrounds persisted into the Cambrian [16], the 'catastrophe' and 'biotic replacement' models both possess supporting lines of evidence and critical questions that remain to be addressed (summarized in [6,15]). Taken globally, there appears to be a transition in assemblage composition between the Ediacaran and the Cambrian [17], though the influence of potential environmental and taphonomic signals, as well as potential diachroneity in the first and last appearances of particular fossil groups, remains to be constrained.

Supporting the 'catastrophe' model, the Shuram and Basal Cambrian Isotope Excursion (BACE) negative $\delta^{13}C$ excursions potentially represent major perturbations to the global carbon cycle [18], may coincide with pulses of extinction [11,19,20], and could reflect a variety of possible kill mechanisms. However, there are still broad uncertainties surrounding the timing, synchroneity and mechanisms responsible for both excursions [21], and thus neither can yet be convincingly linked to putative Ediacaran extinction events [6]. Supporting 'biotic replacement', the latest Ediacaran records an increase in the diversity of metazoan trace and body fossils [22–27] and low diversity of soft-bodied Ediacara biota, potentially representing ecologically stressed communites [13,28,29]. However, a key prediction of this model—that there was an increase in ecosystem engineering impact (EEI) during the latest Ediacaran—has not been tested.

Bioturbation is a crucial ecosystem engineering behaviour in modern marine environments, affecting the oxygenation of the water column [30,31], pore water redox chemistry [32,33], sediment stability [34] and the cycling of marine nutrients [35,36], but different types of trace fossils and burrowing behaviours have varying effects on these processes depending on how the trace maker interacted with the sediment [37–39]. Here, we perform the first robust test of a key tenet of the 'biotic replacement' model by quantifying the intensity of metazoan bioturbation and characterizing ecosystem engineering in trace fossil assemblages from the Ediacaran to Cambrian-aged Nama Group of southern Namibia. These data both provide a test for a key tenet of the 'biotic replacement' model for the extinction of the Ediacara biota and help to establish plausible extinction drivers in a 'biotic replacement' scenario.

## 1.1. Geologic setting

The Nama Group (figure 1) records a greater than 3000 m thick mixed siliciclastic–carbonate succession, deposited into a foreland basin related to convergence along the Damara and Gariep deformational belts [43–45]. The Nama Group is divided into two sub-basins—the northern Zaris Sub-basin and southern Witputs Sub-basin—which are separated by a palaeo-topographic high. Both basins are subdivided into three subgroups: (in ascending stratigraphic order) the Kuibis, the Schwarzrand and the Fish

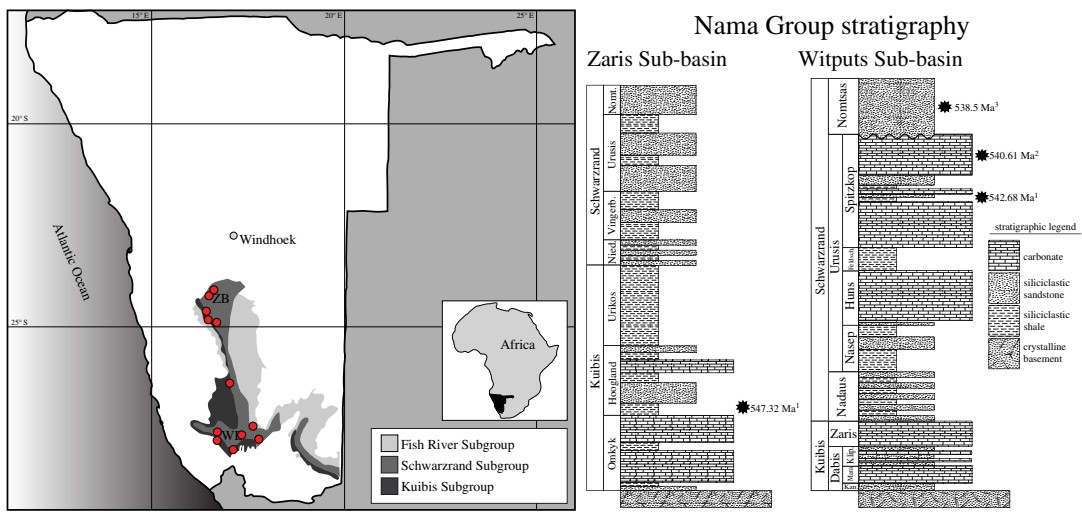

**Figure 1.** Map of Nama Group as it outcrops in southern and central Namibia, southwest Africa, with composite stratigraphy of the Kuibis, Schwarzrand and Fish River Subgroups in the Zaris (ZB) and Witputs (WB) Sub-basins. Erosional unconformity between the Spitzkop and Nomtsas in the Witputs Sub-basin indicated by waved line between the two members. Red circles indicate localities where trace fossil slabs were collected. Composite stratigraphy adapted from Blanco *et al.* [40]. Ash bed dates from (1) Schmitz [41], (2) Narbonne *et al.* [42] and (3) Linnemann *et al.* [2].

River (figure 1). However, precise stratigraphic correlation of subgroups between the two basins is still debated.

In the Witputs Sub-basin, the uppermost valley infill Nomtsas Formation is recognized as Cambrian based on abundant *Treptichnus pedum* [46] and has been dated using U-Pb geochronology at 538.58 ± 0.19 Ma [2]. The underlying Spitzkop Member preserves soft-bodied Ediacara biota as well as *Cloudina* and *Namacalthus* and are thus interpreted to be latest Ediacaran in age [41,42]. The Ediacaran–Cambrian boundary in the Witputs Sub-basin is therefore placed at the erosive unconformity where the base of the Nomtsas Formation cuts down into the Spitzkop Member [43,45]. We note that Linnemann *et al.* [2] instead place the boundary near the top of the Spitzkop Member at Farm Swartpunt, however, given that this stratigraphic placement relies on the identification of *T. pedum* in this section (something which has not been recorded by previous workers), and comes below carbonates containing the skeletonized taxa *Cloudina* and *Namacalathus*, in this study, we follow Saylor *et al.* [45] and Germs [43] in placing the boundary at the unconformity. Ash beds from below the last occurrence of the soft-bodied biota have recently been dated at between 540.095 ± 0.099 Ma and 538.99 ± 0.21 Ma [2]. These new dates place the Ediacaran–Cambrian boundary at 539–538 Ma, depending on which stratigraphic placement is used ([2], their fig. 1).

In the Zaris Sub-basin, a precise stratigraphic location of the Ediacaran–Cambrian boundary has not been determined; an ash bed from the Hoogland Member (Kuibis Subgroup) in the vicinity of the Zebra River Lodge yields a U-Pb zircon age of 547.32 ± 0.65 Ma [41], while the overlying Fish River Subgroup contains *T. pedum* [47]. However, both the disappearance of *Cloudina* in the Urusis Formation [43] and the presence of the tubular taxon *Shaanxilithes* and *Aspidella* in the lower Schwarzrand Subgroup [25] suggest that the Ediacaran–Cambrian boundary probably occurs somewhere in the upper Schwarzrand Subgroup.

## 2. Methods

A total of 57 trace fossil slabs were collected from 12 separate sites across both the Zaris and Witputs Sub-basins, spanning Ediacaran through earliest Cambrian stratigraphy. These slabs typically preserved between tens to hundreds of individual trace fossils. In the Zaris Sub-basin, slabs were collected from five localities in the Niederhagen and Vingerbreek Members and Urusis Formation, while in the Witputs Sub-basin, slabs were collected from seven localities in the Kliphoek, Nasep and Spitzkop Members and the Nomtsas Formation (electronic supplementary material, table S2; figure 1). The trace fossil assemblages collected from near the top of the Spitzkop Member on Farm Swartpunt were sourced from siliciclastic horizons stratigraphically below the Ediacaran–Cambrian boundary inferred by Linnemann *et al.* [2], itself below the unconformity that is more typically interpreted as the boundary [45], and so are unambiguously

Ediacaran, rather than Cambrian in age. Trace fossils on each slab were systematically assessed for biogenicity on the basis of evidence for sediment displacement, consistency of burrow diameter, the absence of frayed and angular burrow terminations and the absence of angled corners (following [48]). Selected slabs were cut and polished to examine putative trace fossils in cross-section. Once the biogenicity of trace fossils was established, ichnotaxa were identified using previous summaries of Ediacaran trace fossils (e.g. [24,49]).

Bioturbation intensity was quantified by digitally point-counting trace fossils on slabs, using $10 \times 10$ cm grids which were superimposed in Adobe Photoshop. Bedding-plane bioturbation percentage was calculated by dividing the number of point-counts by the total number of grid-intersection points (see [50,51]). For slabs larger than $100$ cm$^2$, multiple grids were randomly placed until the majority of the slab was covered, and an average value was taken from all of the grids. For slabs smaller than $100$ cm$^2$, only those as large as an estimated 75% of the grid size were point-counted ($n = 51$ slabs). If a slab contained trace fossils on multiple exposed bedding planes, each bedding plane was counted separately. We use this point-counting method in conjunction with the bedding-plane bioturbation index (BPBI) method [52].

The ecosystem engineering impact (EEI) of trace fossil assemblages was characterized in two ways. First, we applied the 'EEI' indices of Herringshaw *et al.* [37] to each ichnotaxon found on each slab. This method represents a functional bioturbation analysis which provides a measure of ecosystem engineering intensity on the basis of burrow depth, functional group and bioirrigation potential. Second, we used the ecosystem engineering cube method developed by Minter *et al.* [38]; this method focuses on the number of types of ichnological impacts on the sediment by assigning tiering, sediment interaction mechanisms and sediment modification modes. Where the Herringshaw *et al.* [37] EEI method helps characterize the intensity of ecosystem engineering represented by each ichnotaxon, the Minter *et al.* [38] cube method helps characterize the different types of ecosystem engineering behaviours present in trace fossil assemblages. We note that a limitation to the EEI method is a spatial overlap (e.g. the surficial tier and surficial modifiers) and impossible combinations (e.g. the deep tier and epifaunal locomotion) between tiering and functional group [38]. However, these limitations were addressed and are accounted for in Minter *et al.*'s [38] ecosystem engineering cube scheme. Functional group, sediment interaction and sediment modification assignments were given based on supplementary material from Minter *et al.* [38] and additional literature describing the behaviour of Ediacaran and early Cambrian trace fossils (e.g. [23]). Bioturbation percentage and both ecosystem engineering methods results were grouped from the corresponding stratigraphic member in each sub-basin. Bioturbation percentage is given as a distribution of values of each member, and EEI values are given as the range between the absolute minimum and maximum values of all of the ichnogenera found on slabs in the member.

# 3. Results

## 3.1. Trace fossil occurrences

Our dataset records a total of six different ichnotaxa from the Ediacaran and earliest Cambrian portions of the Zaris and Witputs Sub-basins (figure 2): *Helminthopsis*, *Helminthoidichnites*, *Bergaueria*, *Parapsammichnites*, *Treptichnus* and plug-shaped burrows (for descriptions and localities of trace fossils, see electronic supplementary material, S1 and table S2). Trace fossils broadly interpreted as plug-shaped burrows can be classified as either *Conichnus* or *Bergaueria* when cut open. However, due to the inability to distinguish the two ichnogenera on the bedding plane alone when slabs were not able to be cut open (and, although we did not observe this in samples which were cut and polished, some plug-shaped burrows may be poorly preserved *Treptichnus*; see [53]), we group the ichnotaxa together. Surfaces bearing trace fossils are more abundant in the Witputs Sub-basin than in the Zaris Sub-basin, resulting in a large difference in the number of trace fossil slabs analysed from each.

Trace fossil slabs ($n = 13$) from the Zaris Sub-basin were collected from the Niederhagen, Vingerbreek and Urusis Members. Only two ichnotaxa occur on trace fossil slabs: plug-shaped burrows and *Helminthoidichnites* (electronic supplementary material, table S2; figure 3*a*). Trace fossil slabs ($n = 50$) from the Witputs Sub-basin were collected from the Kliphoek, Nasep, Spitzkop and Nomtsas Members. All six ichnotaxa occur in the Witputs Sub-basin (occurrences are detailed in electronic supplementary material, table S2; figure 3*b*).

## 3.2. Bioturbation intensity

All slabs in the Zaris Sub-basin yield low bedding-plane bioturbation intensity (BPBI = 2). The mean bedding-plane bioturbation in the Niederhagen Member is 1.08% ($n = 1$) (electronic supplementary

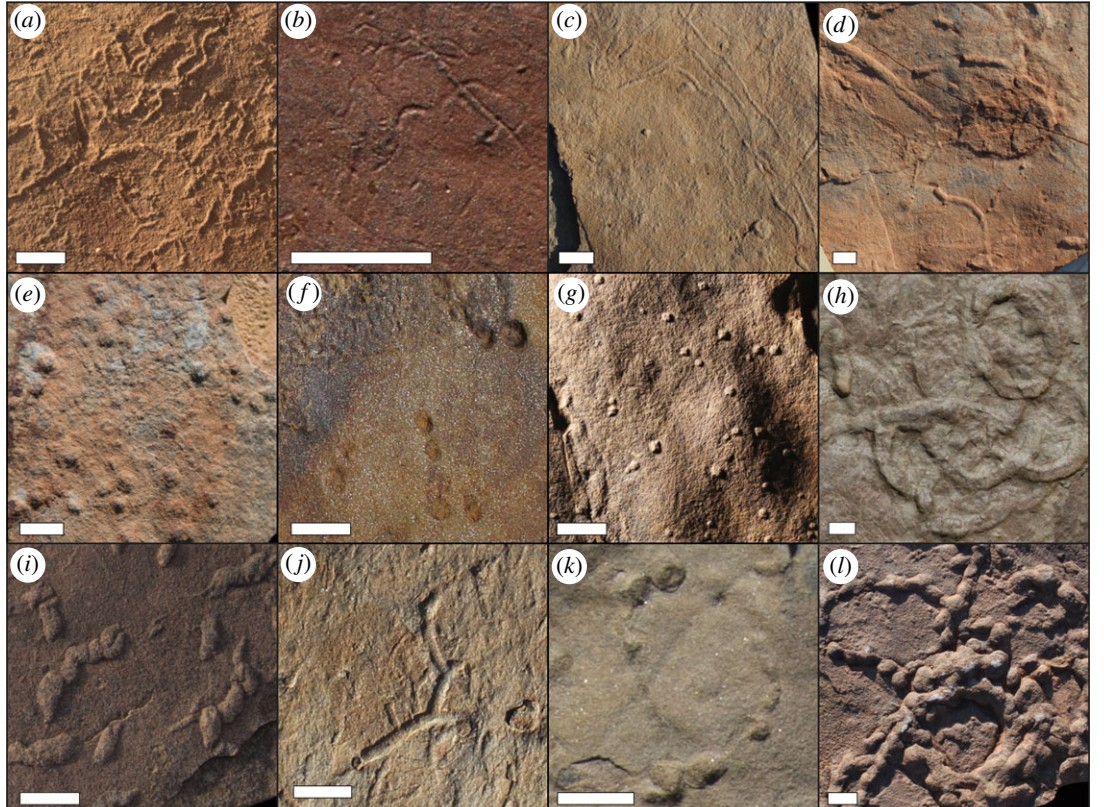

**Figure 2.** Examples of trace fossils which occur in the Nama Group. (*a*) *Helminthoidichnites* from the Canyon Roadhouse locality. Sample 2017-CR-1.06. (*b*) *Helminthopsis* from the Farm Arimas locality. Sample 2017-AR-1.04. (*c*) *Helminthoidichnites* from the Farm Swartpunt locality. Sample TB-16-SP-1.16. (*d*) *Planolites* from the Farm Swartpunt locality. Sample TB-16-SP-1.10. (*e*) Plug-shaped burrows from the Camp Koelkrans locality. Sample 17-FR-2.3. (*f*) *Conichnus* from the Haruchas locality. Note apparent burrow pairing and MISS texture in the upper left corner of the photograph. Sample 2016-HH-1.2. (*g*) Plug-shaped burrows from the Spider Ridge locality. Sample 2016-KK-1.4. (*h*) *Parapsammichnites* from the Camp Koelkrans locality. Sample 17-FR-CK-2. (*i*) Treptichnid-type trace from the Canyon Roadhouse locality. Sample 2017-CR-1.33. (*j*) *Treptichnus* from the Farm Swartpunt locality. Sample TB-16-SP-1.5. (*k*) *Treptichnus* from the Camp Koelkrans locality. Sample 17-FR-1.01. (*l*) *T. pedum* from the Farm Sontaagsbrun locality. Sample 2016-SB-2.31. Detailed descriptions of each trace fossil are given in electronic supplementary material, S1. Scale bars are all 1 cm.

material, table S4; figure 3*a*). Bioturbation increases in the Vingerbreek Member to a mean value of 1.24% (*n* = 9) and increases again into the Urusis at 2.83% (*n* = 1) (electronic supplementary material, table S4; figure 3*a*).

In the Witputs Sub-basin, bedding-plane bioturbation intensity gradually increases towards the Cambrian boundary. The mean percentage value for the Kliphoek Member is 1.94% (*n* = 6), which increases gradually in the Nasep Member to a mean value of 3.34% (*n* = 10) and increases again in the Spitzkop Member to a mean value of 5.61% (*n* = 18) (electronic supplementary material, S4; figure 3*b*). There is a larger shift into the Cambrian Nomtsas Formation to a mean percentage value of 16.0% (*n* = 5) (electronic supplementary material, table S4; figure 3*b*). Trace fossil slabs yield BPBI values of only 2 in the Kliphoek and Nasep Members. The majority (*n* = 15) of trace fossil slabs in the Spitzkop Members yield BPBI = 2, and the remaining three slabs yield BPBI = 3. In the Nomtsas Formation, only one slab yields BPBI = 2, while the rest (*n* = 4) record higher bioturbation (BPBI = 3) (electronic supplementary material, table S4).

## 3.3. Ecosystem engineering

In the Zaris Sub-basin, EEI values from the Niederhagen and Vingerbreek Members yield a small range of EEI values between 5 and 7 due to the sole presence of plug-shaped burrows (figure 3*a*; for EEI assignments, see electronic supplementary material, S5). In the upper Urusis Formation, where *Helminthoidichnites* occurs, EEI values decrease in value to EEI = 3–4 (electronic supplementary material, S2; figure 3*a*). The Zaris ecosystem engineering cubes (figure 4) show that in both the Niederhagen and Vingerbreek Members, only the semi-infaunal compression/gallery biodiffusion cube is occupied by plug-shaped burrows (electronic supplementary material, S6; figure 4*a*). In the

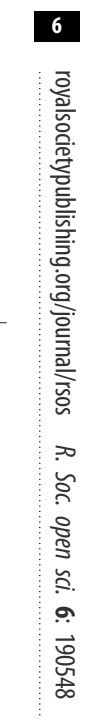

**Figure 3.** Resulting violin plots of point-counted bedding-plane bioturbation percentages, range of EEI values and relative abundance of each trace fossil for the Nama Group. (*a*) Results for the Zaris Sub-basin (from bottom to top) Niederhagen Member, Vingerbreek Member and Urusis Formation. (*b*) Results for the Witputs Sub-basin (from bottom to top) Kliphoek Member, Nasep Member, Spitzkop Member and Nomtsas Formation. Darker grey colours for ichnogenera indicate relatively high EEI values, while lighter grey colours indicate relatively low EEI values. Vertical dashed line on bioturbation percentages plots indicates boundary between BPBI = 2 and BPBI = 3.

Urusis Formation, only the semi-infaunal backfill/conveyor cube is occupied by *Helminthoidichnites* (electronic supplementary material, S6; figure 4*a*).

In the Witputs Sub-basin, Kliphoek Member EEI values are intermediate but small in range at EEI = 5–7 due to the sole presence of plug-shaped burrows (electronic supplementary material, table S2; figure 3*b*). Maximum values and range for the entire Nama Group occur in the Nasep Member (EEI = 3–12; figure 3*b*) and remain constant into the Spitzkop Member. However, the occurrence of *Planolites* and *Parapsammichnites* in the Spitzkop Member result in a higher proportion of high-EEI value trace fossils relative to the Nasep. Finally, in the basal Cambrian Nomtsas Formation, EEI values remain high but contract in range (EEI = 6–12; figure 3*b*). The Witputs ecosystem engineering cubes reflect this pattern. In the Kliphoek Member, where trace fossils are limited to plug-shaped burrows, only the semi-infaunal compression/gallery biodiffusion cube is occupied (electronic supplementary material, table S2; figure 4*b*). In the Nasep and Spitzkop Members, the semi-infaunal compression/backfill cube is also occupied by *Helminthopsis*, *Helminthoidichnites*, *Planolites* and *Parapsammichnites*. Finally, in the Cambrian Nomtsas Formation, only the semi-infaunal compression/gallery biodiffusion cube is occupied by *Treptichnus* (figure 4*b*).

## 4. Discussion

The biotic replacement model for the extinction of the soft-bodied Ediacara biota makes two key predictions: (i) that contemporaneous communities of Ediacara biota are depauperate and ecologically stressed, and (ii) that

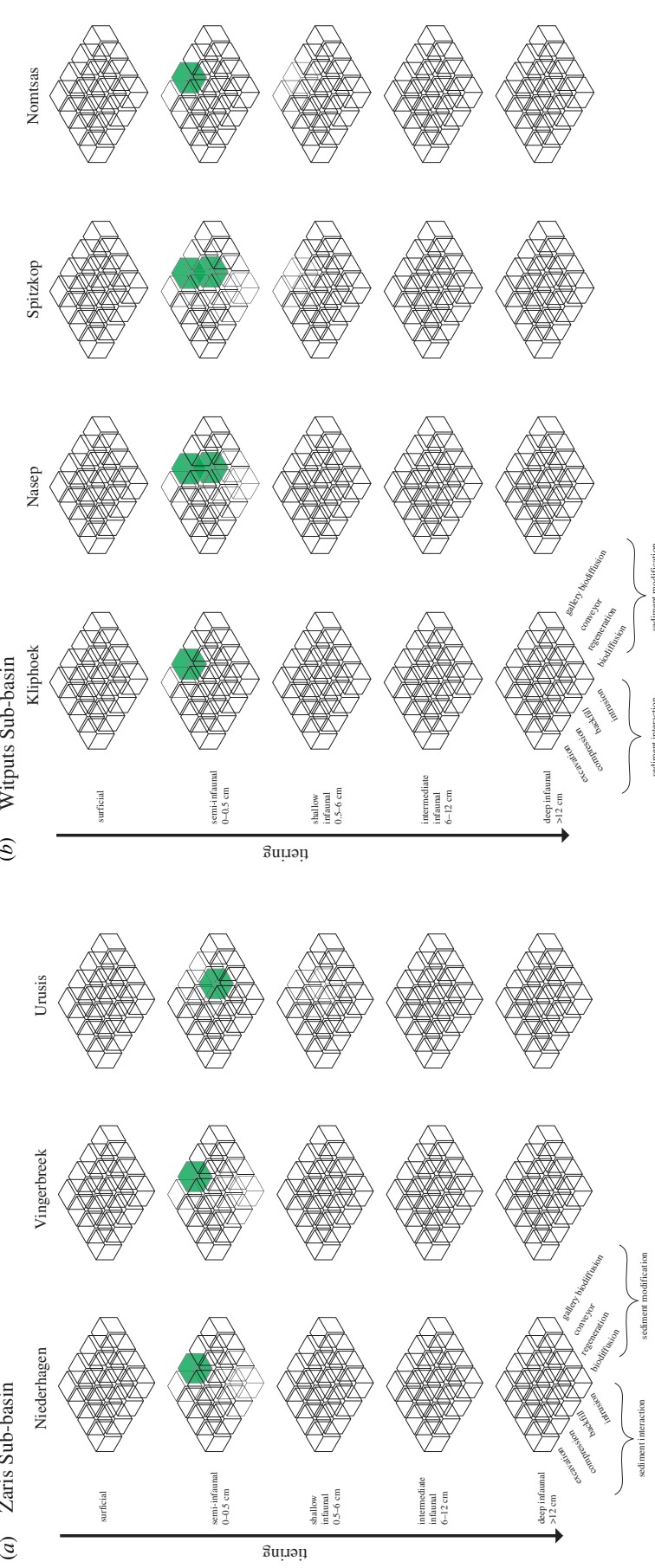

**Figure 4.** Results for the ecosystem engineering cubes as a function of tiering, sediment interaction and sediment modification. Occupied cubes in the (*a*) Zaris Sub-basin and (*b*) Witputs Sub-basin.

Cambrian-type metazoan ecosystem engineering increases in intensity in the latest Ediacaran Nama Group [6,12]. The first prediction of this model is supported by previous studies showing that communities of Ediacara biota in the Nama Group are low-diversity compared with those of the older White Sea assemblage [13,15,54] and are ecologically depauperate, potentially comprising a limited number of life strategies ([28]; although we note that relative abundance community data from the Nama Group are still limited, and thus inferences are preliminary). However, to date, the second prediction of the biotic replacement model—that metazoan ecosystem engineering intensity increases substantially in the latest Ediacaran—had not been demonstrated until now. The results presented here for the Nama Group provide the first robust test of this prediction and help to establish a pattern of ecological changes that reflect the potential effects of ecosystem engineering in latest Ediacaran shallow marine environments.

Our data from the Nama Group illustrate three key features of the latest Ediacaran trace fossil record in Namibia. First, there is a gradual increase in bioturbation intensity from the Ediacaran to Cambrian, with the most substantial increase in bioturbation intensity occurring across the Ediacaran–Cambrian boundary (figure 3). In addition to an increase in the mean bedding-plane bioturbation percentages from the Spitzkop Member to the basal Cambrian Nomtsas Formation, there is also a shift in the distribution of bioturbation intensity values, where the Nomtsas Formation has a greater proportion of high-percentage trace fossil slabs than the Spitzkop Member. However, the Spitzkop Member still contains slabs that yield bioturbation percentage values as high as those in the Nomtsas Formation, indicating that in specific environments, bioturbation intensity was as high in the late Ediacaran as it was at the onset of the Cambrian.

Second, our data illustrate an increase in the complexity of metazoan ecosystem engineering behaviours leading up to the Cambrian. The increase in EEI values observed in the Witputs and Zaris Sub-basins reflects the presence of more complex bioturbation behaviours throughout the Nama Group. The presence of plug-shaped burrows in lower Nama Group members yields mid-value, low-range EEI values (figure 3b), and a limited number of occupied ecosystem engineering cubes (figure 4): one occupied in both the Kliphoek Member (Witputs Sub-basin) and Niederhagen Member (Zaris Sub-basin—figure 4a). Towards the Cambrian, both EEI values and the number of occupied ecosystem engineering cubes increase in the Witputs Sub-basin (figures 3b and 4b), indicating a greater potential for significant environmental modification and the presence of a greater number of ecosystem engineering behaviours. Moreover, diversification in burrowing behaviours is evident throughout the Nama Group. Trace fossils become increasingly more complex in terms of both morphology and inferred feeding strategies. The first trace fossils to appear in stratigraphic succession are plug-shaped burrows, which probably represent metazoans able to adjust their vertical position in the sediment (potentially anthozoan cnidarians—see [25,55]; cf. [56]). By contrast, *Parapsammichnites* records the activity of a bilaterian metazoan possessing a coelom [26] and, along with *Treptichnus*, represents the advent of sediment bulldozing and deposit feeding [23,26,46,57].

Third, our data show that the maximum ecosystem engineering intensities measured in the Nama Group occur below Ediacaran–Cambrian boundary. Specifically, EEIs in the Witputs Sub-basin reach maximum recorded values (EEI = 3–12) and the maximum number of cubes (2) is occupied in the Nasep Member, below the Cambrian Nomtsas Formation (figures 3b and 4b). Trace fossils from the Cambrian Nomtsas Formation yield a further increase in EEI values, but a decrease in the diversity of ecosystem engineering behaviours (represented by a decrease in the number of occupied ecosystem engineering cubes—figure 4). This observed increase in EEIs is a result of the occurrence of treptichnids prior to the appearance of *T. pedum*, while the increase in the number of ecosystem engineering behaviours reflects the increase in ichnodiversity of each stratigraphic member (although we note that it is possible that this reflects an environmental signal, the high diversity trace fossil assemblages may instead reflect some change in taphonomic conditions). Furthermore, the occurrence of other trace fossils which represent similar complex ecosystem engineering behaviours which have been reported from the Nama Group but are not included here, such as *Streptichnus narbonnei* in the Spitzkop Member [22], add to the robustness of these data. Regardless, these results overall suggest that the potential for significant environmental modification due to burrowing was present prior to the appearance of more architecturally complex and larger Cambrian trace fossil makers. This trend in increasing ecosystem engineering prior to the Cambrian is comparable with other Ediacaran–Cambrian sections worldwide, notably in the Ediacaran-aged Blueflower Formation of northwestern Canada where complex burrowing behaviours associated with deposit feeding are present prior to the Ediacaran–Cambrian boundary [23] and in the Chapel Island Formation where the trace fossil record preserves a dramatic increase in behavioural innovation and ecospace occupation from the Ediacaran to the early-middle Cambrian [58]. Overall, the trace fossil record of the Nama Group records an early

increase in ecosystem engineering well before the Ediacaran–Cambrian boundary, which is consistent with the first prediction of the biotic replacement model.

Although our data illustrate an increase in ecosystem engineering in the late Ediacaran, linking specific patterns of environmental change with ecological pressures deleterious to the Ediacara biota has proven difficult [6]. Our analysis of EEI values (of [37]) and ecosystem engineering cubes (of [38]), however, suggest that palaeoenvironmental conditions may have changed in response to diversifying infaunal activity, and thus may help generate hypotheses surrounding the specific mechanisms of biotic replacement. In particular, our analysis shows a latest Ediacaran increase in burrowing behaviours capable of both: (i) altering patterns of nutrient transport and fluid exchange between the substrate and water column, and (ii) altering the rheological properties of the sediment–water interface. In terms of the former, the presence of high-EEI value trace fossils (i.e. *Treptichnus*, *Parapsammichnites*) in the Nasep and Spitzkop Members suggests the early potential for significant sedimentary geochemical change. For example, increased biomixing would have enhanced sulfate reduction in pore waters, while the advent of bioirrigation would have flushed reduced iron and sulfur from the sediment and decreased the formation of iron sulfides thought to be key in preserving the soft-bodied biota [39,59]. Additionally, bioturbation is capable of causing a positive feedback loop by deepening oxygen penetration into the sediment and mixing labile organic matter below the sediment surface, which thus can lead to deeper and more intense metazoan activity and bioturbation [35]. Moreover, disrupting the rheological properties of the sediment surface in this manner could have produced a variety of downstream effects on benthic communities. It has been suggested that the soft-bodied Ediacara biota were evolutionarily adapted to live on microbial matgrounds [60,61]; the diversification of microbial grazing, sediment bulldozing and treptichnid-like metazoan behaviours may have been responsible for clearing large portions of matground in shallow marine environments. Although Buatois *et al.* [16] demonstrate that Ediacaran-style matgrounds persist into the Cambrian, even partial removal of matgrounds may have represented a severe ecological stress for the soft-bodied Ediacara biota, whose bulbous anchoring structures potentially lacked the morphological adaptations to function in less-firm substrates [12]. Future work quantifying the spatial and facies associations between trace fossils, soft-bodied Ediacara biota and microbial mat textures throughout the Ediacaran may provide a test for this model. Additionally, the presence of metazoan deposit-feeders has been shown in some circumstances to have detrimental effects on suspension-feeding species; disrupting the sediment–water interface may inhibit the settlement of larvae and remobilize sediment in a way which 'clogs' the feeding apparatus of suspension-feeding taxa (e.g. [62]). Given that some Ediacara biota have been interpreted as suspension feeders (e.g. [63]), it is possible that increased sediment load in low levels of the water column may have inhibited these species' abilities to obtain nutrients [26]. Although these hypotheses are supported by the trace fossil data, we acknowledge that there are many other possible drivers of biotic replacement and the extinction of the soft-bodied Ediacara biota that would not be recorded in trace fossils—for example, changes to the character of bioavailable organic matter (e.g. [31,64]), competition for resources and the advent of predation (preserved drillholes in biomineralizing *Cloudina*—see [65]—notwithstanding).

Given that the stratigraphic successions preserved in both the Zaris and Witputs Sub-basins record a variety of palaeoenvironments, it is possible that the observed succession in trace fossil assemblages (and associated measures of ecosystem engineering intensity) may be the result of secular changes in facies and/or water depth (e.g. [66,67]). Certainly, many of the horizons hosting more complex trace fossils (in particular, the Nasep and Spitzkop Members) apparently record higher-energy palaeoenvironments, preserving abundant tool marks, groove casts and other evidence for directed transport (electronic supplementary material, S7). However, we note that the most diverse trace fossil assemblages, discovered at the base of the Spitzkop Member (including *Parapsammichnites*, Treptichnid-like traces and a variety of different horizontal burrows), are preserved in thin interbedded sandstones and limestones possessing no evidence for high energy and transport. In addition, trace fossil studies from other late Ediacaran localities have reported Ediacaran-aged *Treptichnus* from both deep (e.g. [23]) and shallow (e.g. [68]) marine environments. Consequently, it is possible that complex infaunal activities with enhanced EEIs may have appeared approximately synchronously in a variety of environments (see also [53]). Untangling evolutionary, environmental and ecological feedbacks is key to robustly testing models of biotic extinction and replacement, and to resolving long-withstanding questions regarding the development of Phanerozoic-style metazoan ecosystems.

In summary, our study provides the first robust test for a key prediction of the biotic replacement model for the Ediacaran–Cambrian transition—namely, an increase in metazoan ecosystem engineering in the latest Ediacaran. Our trace fossil data from the Nama Group of southern Namibia illustrate a gradual

increase in bioturbation intensity throughout the Nama Group, but a dramatic increase in the EEI of trace fossils in the latest Ediacaran. These increases in diversity and EEIs pre-date or are at least contemporaneous with the appearance of low-diversity and potentially ecologically stressed communities of soft-bodied Ediacara biota in the same basin [25,28] and are thus consistent with the biotic replacement model. Direct causal links between the appearance of these new infaunal behaviours and the extinction of the Ediacara biota are still lacking, in part due to the rarity of successions hosting both soft-bodied macrobiota and abundant trace fossils. However, our approach provides the ichnological and stratigraphic basis for hypotheses that can be tested with future ecological data and geochemical evidence. We also note that the trace fossil data presented here are not mutually exclusive of a 'catastrophe'-type scenario involving abiotically induced environmental changes (e.g. [20]). Throughout geologic time, periods of evolutionary innovation have been associated with environmental fluctuations in oxygen availability (e.g. [69]). Emerging geochemical data suggest the end Ediacaran was defined by local redox instability in shallow-water strata [70], reflecting major transitions in the state of global marine redox coinciding with both the end of the Shuram excursion (*ca* 550 Ma) and the Ediacaran–Cambrian boundary [71–73]. Therefore, the trace fossil record of the Nama Group may reflect innovation in ecosystem engineering and radiation of bilateria in response to 'catastrophe'-type environmental events related to expanding ocean anoxia (or some other environmental perturbation) that ultimately contributed to the demise of the Ediacara biota.

Finally, our findings highlight that further work is needed to spatially constrain Nama Group trace fossils in the context of the contemporaneous soft-bodied Ediacara biota, to understand changes in bioturbation during the Avalon and White Sea assemblages and to determine a precise biogeochemical mechanism(s) associated with bioturbation that could have led to, or arisen from, environmental changes. Despite this, our study provides a link between ecosystem engineering and the extinction of the Ediacara biota, provides insights into the evolution of bioturbation prior to the Cambrian and illustrates that early metazoans were capable of achieving levels of ecosystem engineering approaching those that appear in the earliest Cambrian by the latest Ediacaran.

Ethics. Permission to collect samples was granted to all authors by the following landowners: Patricia Craven and Herman Kinghorn for Farm Arimas, Lothar Gessert for Farm Swartpunt, Leez Hovelmann at Gondwana Canyon Park for Camp Koelkrans, Barra and Lilian Viljoen for Farm Hansburg, and Gwen Finis at Farm Berghoek. Permission was not required at the Canyon Roadhouse locality. At all other farms, permission was obtained at the farmhouse before conducting fieldwork. Research permits were obtained from the National Heritage Council (permit no. 04/2017). This work was done in collaboration with the Ministry of Mines and Energy in Windhoek, Namibia.
Data accessibility. Datasets supporting this article have been uploaded as part of the electronic supplementary material.
Authors' contributions. A.T.C., C.G.K., M.L. and S.A.F.D. designed the research. All the authors performed the fieldwork and helped collect trace fossils and slabs for analysis. A.T.C., C.G.K., T.H.B., M.L. and S.A.F.D. wrote the manuscript. All the authors gave final approval for publication.
Competing interests. We declare we have no competing interests.
Funding. This research was supported by a Paleontological Society Student Research Grant to A.T.C. S.A.F.D. acknowledges generous funding from National Geographic (grant no. 9968-16) and a Paleontological Society Arthur James Boucot Award. M.L. acknowledges funding from NSERC Discovery Grant (grant no. RGPIN 435402). R.A.R. was supported by NSF (grant nos. DEB 1331980 and PLR 134175) to N. Smith.
Acknowledgements. All the authors would like to extend heartfelt thanks to Barra and Lilian Viljoen, Danie Loots, Patricia Craven, Herman Kinghorn and Lothar Gessert for access to farms and fossil sections. We also thank the Geological Survey of Namibia, in particular Charlie Hoffmann and Roger Swart, for logistical help in conducting fieldwork. We thank two anonymous reviewers for thoughtful comments. Lastly, we extend sincere thanks to Gerard Germs and John Almond for helping us find the trace fossil sites in the Nasep and basal Spitzkop Members.

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
