## [Reviewer comments · Royal Society Open Science]

Review History

RSOS-190548.R0 (Original submission)

Review form: Reviewer 1 (Andrey Yurevich Zhuravlev)

Is the manuscript scientifically sound in its present form?

No

Are the interpretations and conclusions justified by the results?

No

Is the language acceptable?

Yes

Is it clear how to access all supporting data?

Yes

Do you have any ethical concerns with this paper?

No

Have you any concerns about statistical analyses in this paper?

I do not feel qualified to assess the statistics

Recommendation?

Major revision is needed (please make suggestions in comments)

Comments to the Author(s)

Review of RSOS-190548: Increase in metazoan ecosystem engineering prior to the Ediacaran-Cambrian boundary in the Nama Group, Namibia by A. Cribb et al.

The manuscript by Alison T. Cribb and his colleagues discuss a many-sided problem of the Ediacaran-Cambrian transition heated by current achievements in geochemistry and palaeontology of this important interval in the Earth history. At the same time, new data show a mosaic pattern of environmental and biotic processes during this transition in different regions. Although the authors' data and interpretation are reliable in the limits of the Namibian basins, the thoughts that their model is applicable to global events can be disputed (see below). A clear timescale has to be adjusted by the authors to their study including each plot to make the discussion understandable.

GENERAL COMMENTS:

Lines 48 and below: The terminal Neoproterozoic Ediacaran Period is dated by 635–541 ± 1.0 Ma according to the International Chronostratigraphic Chart issued by International Commission on Stratigraphy in 2018 (www.stratigraphy.org). Thus, the authors' data frame of 635–538.5 Ma contradicts this figure and turns the entire problem under discussion to the shady zone of the age misinterpretations which, unfortunately, is among the key points of the reliability of the entire study: What events we actually discuss if these events occur 541–538.5 Ma--the late Ediacaran or the early Cambrian?

Lines 107-112: Moreover, if the Ediacaran-Cambrian boundary is 'placed at the erosive unconformity' between strata dated at 541.61 Ma and 538.5 Ma, how can the authors prove that nothing has happened during 3 my gap embracing the very time of events under discussion? If the biotic replacement model discussed by the authors is verifiable at the Ediacaran-Cambrian successions of Namibia due to the presence of a serious time gap?

Lines 299-302: For instance, if 'the trace fossil record of the Nama Group... illustrates Cambrian-equivalent levels of metazoan ecosystem engineering in specific horizons and palaeoenvironments in the Ediacaran', the authors' model is 'consistent with the first prediction of the biotic replacement model'. However, if these horizons are already Cambrian the model is inconsistent.

Fig. 3. This key figure is absolutely unclear: Here the Nomtsas Member has to be shown as a chronological subdivision above the Spitzkopf Member (Fig. 1 provides the sedimentological record). Now, it is impossible to understand to what a member the Bioturbation Intensity, EEI values, and Ichnogenera Relative Abundance plots do correspond.

MINOR CORRECTIONS

Lines 117-118: A record of *Shaanxilithes* by Darroch et al. (2016, figs 2c-d, 6) is erroneous. This tiny chambered form is a palaeopascichnid – another terminal Ediacaran index-fossil.

Line 250: record not recd

Review form: Reviewer 2 (Luis A. Buatois)

Is the manuscript scientifically sound in its present form?

Yes

Are the interpretations and conclusions justified by the results?

Yes

Is the language acceptable?

Yes

Is it clear how to access all supporting data?

Yes

Do you have any ethical concerns with this paper?

No

Have you any concerns about statistical analyses in this paper?

I do not feel qualified to assess the statistics

Recommendation?

Accept with minor revision (please list in comments)

Comments to the Author(s)

The role of bioturbation during the Ediacaran-Cambrian transition, including its significance in ecosystem engineering, is a central issue for our understanding of the history of life. In addition, the thick and continuous terminal Ediacaran deposits of Namibia are instrumental to address this topic. The vast majority of studies dealing with the impact of bioturbation during the Ediacaran-Cambrian transition are simply based on measuring density of trace fossils in both cross-section and bedding-plane views. The authors should be commended for not falling in this trap and approaching this topic employing more sophisticated tools. With a little bit of additional work, this study will be a valuable addition to the literature on the topic.

My only criticism in this regard is the use of the Ecosystem Engineering Impact index of Herringshaw et al. (2017). This index encompasses several metrics, but without actually articulating the different elements in a consistent fashion. For example, Herringshaw et al. (2017) adopted the functional groups of Solan and co-workers without any adjustments. This is problematic because adjustments are needed for applying this scheme to the fossil record when dealing with tiering structure to avoid conceptual overlap. This was discussed by Minter et al. (2016, p. 22), who stated "We omit the categories of epifaunal bioturbators and surficial modifiers (Solan and Wigham 2005) because of their spatial constraints, being covered potentially by any of the other four means of sediment reworking in the epifaunal and semi-infaunal tier". This overlap compromises the evaluation of the intensity of ecosystem engineering represented by each ichnotaxon in the Ecosystem Engineering Impact approach and as a result the values obtained in this way may be rather arbitrary. My suggestion here is that the authors may well dispense of the Ecosystem Engineering Impact approach and just use the scheme put forward by Minter and co-workers, which covers the same ground on a more consistent fashion.

Also, it may be advisable not to include in the analysis those forms which have not been assigned to an ichnotaxon because the behavioural significance of these may be unclear. In any case, looking at the photographs, I would tend to think that those trace fossils illustrated in Fig. 2A may belong in Helminthoidichnites, the one in Fig. 2B may be referred to as a treptichnid (as the one in Fig. 2E, as indicated by the authors), and the one in Fig. D may be tentatively called

Helminthopsis. The one in Fig. 2C is simply too fragmentary to say. The ones in Fig. 2F may be plug-shaped burrows as indicated by the authors, but also a preservational variant of a treptichnid (cutting the sample may resolve this issue). The one in Fig. 2G is really unconvincing and I am not even sure if it is a trace fossil. The one assigned to Planolites in Fig. 2I is unconvincing, I would place it in Helminthoidichnites. The one assigned to Streptichnus in Fig. 2K (note that the label is gone) is not convincing either, and it is too partially preserved to suggest an alternative name. In short, some adjustments are needed on this front and some of these will translate into changes in the evaluation of ecosystem engineering.

Finally, the author may wish to cite a recent paper that uses a similar approach to assess ecospace utilization in the interval of the Chapel Island Formation attributed to the *Treptichnus pedum* zone, comparing with the underlying strata in the same area. The reference is:

Laing, B.A., Mángano, M.G., Buatois, L.A., Narbonne, G.M. & Gougeon, R.C. 2019. A protracted Ediacaran–Cambrian transition: an ichnologic ecospace analysis of the Fortunian in Newfoundland, Canada. *Geological Magazine*.

A few additional comments:

Lines 57-58: some words are missing in this sentence

Line 62: for the reader unfamiliar with the recent debate, the first part of the sentence may be hard to understand. I suggest listing the three original hypotheses and then discard the taphonomic one citing the Buatois et al. (2014) paper.

Lines 78-79: I think that the authors meant to cite Mángano & Buatois (2014) instead of 2015 here.

Line 108: “Spitskop” x “Spitzkop” (and elsewhere in the text and in Figure 1)

Line 124: “trace fossils” x “traces”

Line 158: Using Carbone and Narbonne (2014) here may be complicated because trace fossils in that study are mostly classified in open nomenclature rather than simply assigned to specific ichnotaxa.

Line 177: “Trace fossil assemblages present in slabs (n=16) from the Zaris Subbasin were of comparatively low diversity....” x “Trace fossil slabs (n=16) from the Zaris Subbasin were comparatively low diversity....”.

Lines 179-180: “In contrast, trace fossil assemblages present in slabs (n=61) from the Witputs Subbasin were of relatively high diversity....” x “In contrast, trace fossil slabs (n=61) from the Witputs Subbasin were relatively high diversity....”

Line 276: “cf.” x “c.f.”

Lines 291-293: It would be useful to back up this statement with some references or more precise facies information.

Line 335: This point is also made in Buatois et al. (2018; *Scientific Reports*), so this paper should be cited here.

Lines 353-354: The succession studied in Gehling et al. (2001) is actually relatively shallow (see, for example, some of the classic papers by Myrow).

Lines 389-391: I strongly suggest toning down this last part of the last sentence. Cambrian levels of ecosystem engineering (particularly since Stage 2) are much higher than terminal Ediacaran levels. There is a fair amount of studies published recently assessing, for example, the increase of bioturbation intensity since Cambrian Age onwards; see Mángano and Buatois, 2014, *Proc B*; Gougeon et al., 2018; *Nature Communications*).

The organizations of the reference list is not consistent with the style of citation through the text.

Decision letter (RSOS-190548.R0)

15-Jul-2019

Dear Ms Cribb,

The editors assigned to your paper ("Increase in metazoan ecosystem engineering prior to the Ediacaran-Cambrian boundary in the Nama Group, Namibia") have now received comments from reviewers. We would like you to revise your paper in accordance with the referee and Associate Editor suggestions which can be found below (not including confidential reports to the Editor). Please note this decision does not guarantee eventual acceptance.

Please submit a copy of your revised paper before 07-Aug-2019. Please note that the revision deadline will expire at 00.00am on this date. If we do not hear from you within this time then it will be assumed that the paper has been withdrawn. In exceptional circumstances, extensions may be possible if agreed with the Editorial Office in advance. We do not allow multiple rounds of revision so we urge you to make every effort to fully address all of the comments at this stage. If deemed necessary by the Editors, your manuscript will be sent back to one or more of the original reviewers for assessment. If the original reviewers are not available, we may invite new reviewers.

- Data accessibility

If you wish to submit your supporting data or code to Dryad (<http://datadryad.org/>), or modify your current submission to dryad, please use the following link:
<http://datadryad.org/submit?journalID=RSOS&manu=RSOS-190548>

- **Competing interests**

- **Authors' contributions**

- **Acknowledgements**

- **Funding statement**

Kind regards,

Andrew Dunn

on behalf of Professor Rachel Wood (Associate Editor) and Jon Blundy (Subject Editor)
openscience@royalsociety.org

Comments to Author:

Reviewers' Comments to Author:

Reviewer: 1

Comments to the Author(s)

Review of RSOS-190548: Increase in metazoan ecosystem engineering prior to the Ediacaran-Cambrian boundary in the Nama Group, Namibia by A. Cribb et al.

The manuscript by Alison T. Cribb and his colleagues discuss a many-sided problem of the Ediacaran-Cambrian transition heated by current achievements in geochemistry and palaeontology of this important interval in the Earth history. At the same time, new data show a mosaic pattern of environmental and biotic processes during this transition in different regions. Although the authors' data and interpretation are reliable in the limits of the Namibian basins, the thoughts that their model is applicable to global events can be disputed (see below). A clear timescale has to be adjusted by the authors to their study including each plot to make the discussion understandable.

GENERAL COMMENTS:

Lines 48 and below: The terminal Neoproterozoic Ediacaran Period is dated by 635–541 ± 1.0 Ma according to the International Chronostratigraphic Chart issued by International Commission on Stratigraphy in 2018 (www.stratigraphy.org). Thus, the authors' data frame of 635–538.5 Ma contradicts this figure and turns the entire problem under discussion to the shady zone of the age misinterpretations which, unfortunately, is among the key points of the reliability of the entire study: What events we actually discuss if these events occur 541–538.5 Ma--the late Ediacaran or the early Cambrian?

Lines 107-112: Moreover, if the Ediacaran-Cambrian boundary is 'placed at the erosive unconformity' between strata dated at 541.61 Ma and 538.5 Ma, how can the authors prove that nothing has happened during 3 my gap embracing the very time of events under discussion? If the biotic replacement model discussed by the authors is verifiable at the Ediacaran-Cambrian successions of Namibia due to the presence of a serious time gap?

Lines 299-302: For instance, if 'the trace fossil record of the Nama Group... illustrates Cambrian-equivalent levels of metazoan ecosystem engineering in specific horizons and palaeoenvironments in the Ediacaran', the authors' model is 'consistent with the first prediction of the biotic replacement model'. However, if these horizons are already Cambrian the model is inconsistent.

Fig. 3. This key figure is absolutely unclear: Here the Nomtsas Member has to be shown as a chronological subdivision above the Spitzkopf Member (Fig. 1 provides the sedimentological record). Now, it is impossible to understand to what a member the Bioturbation Intensity, EEI values, and Ichnogenera Relative Abundance plots do correspond.

MINOR CORRECTIONS

Lines 117-118: A record of *Shaanxilithes* by Darroch et al. (2016, figs 2c-d, 6) is erroneous. This tiny chambered form is a palaeopascichnid – another terminal Ediacaran index-fossil.

Line 250: record not recd

Reviewer: 2

Comments to the Author(s)

The role of bioturbation during the Ediacaran-Cambrian transition, including its significance in ecosystem engineering, is a central issue for our understanding of the history of life. In addition, the thick and continuous terminal Ediacaran deposits of Namibia are instrumental to address this

topic. The vast majority of studies dealing with the impact of bioturbation during the Ediacaran-Cambrian transition are simply based on measuring density of trace fossils in both cross-section and bedding-plane views. The authors should be commended for not falling in this trap and approaching this topic employing more sophisticated tools. With a little bit of additional work, this study will be a valuable addition to the literature on the topic.

My only criticism in this regard is the use of the Ecosystem Engineering Impact index of Herringshaw et al. (2017). This index encompasses several metrics, but without actually articulating the different elements in a consistent fashion. For example, Herringshaw et al. (2017) adopted the functional groups of Solan and co-workers without any adjustments. This is problematic because adjustments are needed for applying this scheme to the fossil record when dealing with tiering structure to avoid conceptual overlap. This was discussed by Minter et al. (2016, p. 22), who stated "We omit the categories of epifaunal bioturbators and surficial modifiers (Solan and Wigham 2005) because of their spatial constraints, being covered potentially by any of the other four means of sediment reworking in the epifaunal and semi-infaunal tier". This overlap compromises the evaluation of the intensity of ecosystem engineering represented by each ichnotaxon in the Ecosystem Engineering Impact approach and as a result the values obtained in this way may be rather arbitrary. My suggestion here is that the authors may well dispense of the Ecosystem Engineering Impact approach and just use the scheme put forward by Minter and co-workers, which covers the same ground on a more consistent fashion.

Also, it may be advisable not to include in the analysis those forms which have not been assigned to an ichnotaxon because the behavioural significance of these may be unclear. In any case, looking at the photographs, I would tend to think that those trace fossils illustrated in Fig. 2A may belong in Helminthoidichnites, the one in Fig. 2B may be referred to as a treptichnid (as the one in Fig. 2E, as indicated by the authors), and the one in Fig. D may be tentatively called Helminthopsis. The one in Fig. 2C is simply too fragmentary to say. The ones in Fig. 2F may be plug-shaped burrows as indicated by the authors, but also a preservational variant of a treptichnid (cutting the sample may resolve this issue). The one in Fig. 2G is really unconvincing and I am not even sure if it is a trace fossil. The one assigned to Planolites in Fig. 2I is unconvincing, I would place it in Helminthoidichnites. The one assigned to Streptichnus in Fig. 2K (note that the label is gone) is not convincing either, and it is too partially preserved to suggest an alternative name. In short, some adjustments are needed on this front and some of these will translate into changes in the evaluation of ecosystem engineering.

Finally, the author may wish to cite a recent paper that uses a similar approach to assess ecospace utilization in the interval of the Chapel Island Formation attributed to the *Treptichnus pedum* zone, comparing with the underlying strata in the same area. The reference is:

Laing, B.A., Mángano, M.G., Buatois, L.A., Narbonne, G.M. & Gougeon, R.C. 2019. A protracted Ediacaran-Cambrian transition: an ichnologic ecospace analysis of the Fortunian in Newfoundland, Canada. *Geological Magazine*.

A few additional comments:

Lines 57-58: some words are missing in this sentence

Line 62: for the reader unfamiliar with the recent debate, the first part of the sentence may be hard to understand. I suggest listing the three original hypotheses and then discard the taphonomic one citing the Buatois et al. (2014) paper.

Lines 78-79: I think that the authors meant to cite Mángano & Buatois (2014) instead of 2015 here.

Line 108: "Spitskop" x "Spitzkop" (and elsewhere in the text and in Figure 1)

Line 124: "trace fossils" x "traces"

Line 158: Using Carbone and Narbonne (2014) here may be complicated because trace fossils in that study are mostly classified in open nomenclature rather than simply assigned to specific ichnotaxa.

Line 177: "Trace fossil assemblages present in slabs (n=16) from the Zaris Subbasin were of comparatively low diversity...." x "Trace fossil slabs (n=16) from the Zaris Subbasin were comparatively low diversity....".

Lines 179-180: "In contrast, trace fossil assemblages present in slabs (n=61) from the Witputs Subbasin were of relatively high diversity...." x "In contrast, trace fossil slabs (n=61) from the Witputs Subbasin were relatively high diversity...."

Line 276: "cf." x "c.f."

Lines 291-293: It would be useful to back up this statement with some references or more precise facies information.

Line 335: This point is also made in Buatois et al. (2018; Scientific Reports), so this paper should be cited here.

Lines 353-354: The succession studied in Gehling et al. (2001) is actually relatively shallow (see, for example, some of the classic papers by Myrow).

Lines 389-391: I strongly suggest toning down this last part of the last sentence. Cambrian levels of ecosystem engineering (particularly since Stage 2) are much higher than terminal Ediacaran levels. There is a fair amount of studies published recently assessing, for example, the increase of bioturbation intensity since Cambrian Age onwards; see Mángano and Buatois, 2014, Proc B; Gougeon et al., 2018; Nature Communications).

The organizations of the reference list is not consistent with the style of citation through the text.

Author's Response to Decision Letter for (RSOS-190548.R0)

See Appendix A.

Decision letter (RSOS-190548.R1)

23-Aug-2019

Dear Ms Cribb,

I am pleased to inform you that your manuscript entitled "Increase in metazoan ecosystem engineering prior to the Ediacaran-Cambrian boundary in the Nama Group, Namibia" is now accepted for publication in Royal Society Open Science.

on behalf of Professor Rachel Wood (Associate Editor) and Jon Blundy (Subject Editor)
openscience@royalsociety.org

Appendix A

Reviewer 1

- 1) “Although the authors’ data and interpretation are reliable in the limits of the Namibian basins, the thoughts that their model is applicable to global events can be disputed (see below). A clear timescale has to be adjusted by the authors to their study including each plot to make the discussion understandable.”

CHANGES MADE: On the point of geochronology we agree with the reviewer here – this is a tricky point. Although the official ICS chart states that the Ediacaran Period spans ~635-541 Ma, new and revised dates from several Ediacaran-Cambrian boundary sections (including Namibia) are producing younger dates for the upper boundary ~539 Ma - see also our response to point 2 below. However, this does not mean that our studied sections are Cambrian; the ICS timescale is guided by the most recent and accurate radiometric ages, not the other way around. Thus, although the revised dates for the uppermost Spitzkop Member now fall within what the 2018 ICS chart would deem Cambrian, these sections are still incontrovertibly Ediacaran – containing a number of key Late Ediacaran index fossils, such as *Swartpuntia*, *Pteridinium*, *Cloudina*, and *Namacalathus*. In fact, a ~539 Ma date for the Ediacaran-Cambrian boundary is emerging from several other sections worldwide, particularly based on work by Linnemann et al. (2019), and so we expect this date to be adopted by the ISC in the near future.

However, we agree that this is a key point that may be lost on some readers, and so we have made a number of adjustments to the text and figures that makes these points clear. We detail these in our responses to points 2, 5 and 6 below.

- 2) “The terminal Neoproterozoic Ediacaran Period is dated by 635–541±1.0 Ma according to the International Chronostratigraphic Chart issued by International Commission on Stratigraphy in 2018 (www.stratigraphy.org). Thus, the authors’ data frame of 635–538.5 Ma contradicts this figure and turns the entire problem under discussion to the shady zone of the age misinterpretations which, unfortunately, is among the key points of the reliability of the entire study: What events we actually discuss if these events occur 541–538.5 Ma—the late Ediacaran or the early Cambrian?”

CHANGES MADE: The dates used in this study (in particular, the 538-539 Ma date measured from the top of the Spitzkop Member) are based on work by Linnemann et al. (2019) who provide the most recent geochronological dates for the Nama Group. Although Linnemann et al. (2019) do break with tradition in placing the Ediacaran-Cambrian boundary within the exposures at Farm Swartpunt (whereas most workers in the region actually place it above the top of the koppe – the disagreement hinges on the identification of *T. pedum*), our trace fossil slabs were collected from *below* this top ash bed. Thus, even given the new dates and new boundary, our slabs have to be considered Ediacaran in age. Offering further support for this, our sampled fossil horizons are bracketed above and below by carbonates containing the skeletonized Ediacaran fossils *Namacalathus* and *Cloudina* (we note that although ‘Cloudinids’ have been reported in Cambrian horizons in China, they were not assigned to *Cloudina* specifically. *Namacalathus*, to the best of our knowledge, is exclusively Ediacaran), occur among and

shortly above horizons containing the soft-bodied taxa *Swartpuntia* and *Pteridinium*. In addition (but just as importantly), the new ~538-539 Ma date for the Ediacaran-Cambrian boundary produced by Linnemann et al. (2019) (and used here) lines up with other recent geochronological work done elsewhere in the world – see e.g. Tsukui et al. (2016) for a ~539 Ma date for this boundary in China. There is thus no reason at all to believe that our fossils are Cambrian, and a new date for the Ediacaran-Cambrian boundary will slowly filter its way into the ICS chart once it has been re-established in other key sections worldwide.

We do, however, agree with the reviewer that this point needs to be clear in the manuscript, and thus we have added the following text to the Geological Setting and Methods sections which should make these points clear.

“The Ediacaran-Cambrian boundary in the Witputs Subbasin is therefore placed at the erosive unconformity where the base of the Nomtsas Formation cuts down into the Spitzkop Member (Saylor et al., 1995; Germs, 1983). We note that Linnemann et al. (2019) instead place the boundary near the top of the Spitzkop Member at Farm Swartpunt, however, given that this stratigraphic placement relies on the identification of *T. pedum* in this section (something which has not been recorded by previous workers), and comes below carbonates containing the skeletonized taxa *Cloudina* and *Namacalathus*, in this study we follow Saylor et al. (1995) and Germs (1983) in placing the boundary at the unconformity.” (lines 117-127).

And:

“The trace fossil assemblages collected from near the top of the Spitzkop Member on Farm Swartpunt were sourced from siliciclastic horizons stratigraphically below the Ediacaran-Cambrian boundary inferred by Linnemann et al. (2019), itself below the unconformity that is ore typically interpreted as the boundary (Saylor et al., 1995), and so are unambiguously Ediacaran, rather than Cambrian in age.” (lines 145-148).

- 3) “Moreover, if the Ediacaran-Cambrian boundary is ‘placed at the erosive unconformity’ between strata dated at 541.61 Ma and 538.5 Ma, how can the authors prove that nothing has happened during 3 my gap embracing the very time of events under discussion?”

CHANGES NOT MADE: We suspect that there may be a fundamental misunderstanding on the part of the reviewer here – the point of our study was to quantify the ecosystem engineering impacts (EEIs) of trace fossils throughout the Ediacaran and Cambrian portions of the Nama Group and to test a key tenet of the ‘biotic replacement’ hypothesis. Our data illustrate that the EEIs of trace fossils increase prior to the Ediacaran-Cambrian boundary (specifically, in both the Nasep and basal Spitzkop Members), and thus that the environment was being modified prior to the appearance of a Cambrian-type biota. To be clear...we draw no conclusions on the nature of Ediacaran-Cambrian boundary itself, and do not in any way dismiss the ‘catastrophe’ model (and, in fact, we dedicate an entire paragraph beginning line 387 explaining how our data do, and do not, fit into the ‘biotic replacement’ vs. ‘catastrophe’ debate). Something catastrophic

may well occur at the Ediacaran-Cambrian boundary in Namibia during the depositional hiatus, but this does not in any way impact our data that show an increase in metazoan ecosystem engineering millions of years beforehand.

- 4) “If the biotic replacement model discussed by the authors is verifiable at the Ediacaran-Cambrian successions of Namibia due to the presence of a serious time gap?”

CHANGES NOT MADE: As above, this comment seems to stem from a misunderstanding of the text – our study clearly articulates that our results support a key prediction of the ‘biotic replacement’ hypothesis, but also that, “the trace fossil data presented here are not mutually exclusive of a ‘catastrophe’-type scenario involving abiotically-induced environmental changes” (lines 399-401). Thus, our study is not designed to provide a concrete test between the two models, but rather to test a key tenet of one model and does not in any way dismiss the other.

- 5) “For instance, if ‘the trace fossil record of the Nama Group... illustrates Cambrian-equivalent levels of metazoan ecosystem engineering in specific horizons and palaeoenvironments in the Ediacaran’, the authors’ model is ‘consistent with the first prediction of the biotic replacement model’. However, if these horizons are already Cambrian the model is inconsistent.”

CHANGES MADE: As per our response to comment #2, all the fossil horizons collected and quantified in this study are incontrovertibly Ediacaran, rather than Cambrian, and we now make this clear with the addition of the following text:

“The Ediacaran-Cambrian boundary in the Witputs Subbasin is therefore placed at the erosive unconformity where the base of the Nomtsas Formation cuts down into the Spitzkop Member (Saylor et al., 1995; Germs, 1983). We note that Linnemann et al. (2019) instead place the boundary near the top of the Spitzkop Member at Farm Swartpunt, however, given that this stratigraphic placement relies on the identification of *T. pedum* in this section (something which has not been recorded by previous workers), and comes below carbonates containing the skeletonized taxa *Cloudina* and *Namacalathus*, in this study we follow Saylor et al. (1995) and Germs (1983) in placing the boundary at the unconformity.” (lines 117-127).

And:

“The trace fossil assemblages collected from near the top of the Spitzkop Member on Farm Swartpunt were sourced from siliciclastic horizons stratigraphically below the Ediacaran-Cambrian boundary inferred by Linnemann et al. (2019), itself below the unconformity that is ore typically interpreted as the boundary (Saylor et al., 1995), and so are unambiguously Ediacaran, rather than Cambrian in age.” (lines 145-148).

- 6) “[Figure 3] is absolutely unclear: Here the Nomtsas Member has to be shown as a chronological subdivision above the Spitzkop Member (Fig. 1 provides the sedimentological record). Now, it is impossible to understand to what a member the

Bioturbation Intensity, EEI values, and Ichnogenera Relative Abundance plots do correspond.”

CHANGES MADE: We have changed the composite stratigraphy in Figure 1 and Figure 3 to make this relationship clearer. Instead of showing the Nomtsas Formation downcutting far into the Spitzkop Member, we now show this as a wavy line between the two members.

- 7) “A record of *Shaanxilithes* by Darroch et al. (2016, figs 2c-d, 6) is erroneous. This tiny chambered form is a palaeopascichnid—another terminal Ediacaran index-fossil.”

CHANGES NOT MADE: We respectfully disagree with the Reviewer here – having discovered this site, collected the material, and performed a detailed analysis of these fossils, Darroch et al. (2016) concluded that these specimens differed in several important regards from paleopascichnids (which are most likely a form of chambered protist, rather than a tube-dwelling metazoan – see Antcliffe et al., 2011). Key differences between the fossils identified by Darroch et al. as *Shaanxilithes* and *Paleopascichnus* are: 1) the maintenance of a consistent tube width; 2) the absence of bifurcations in tubes; 3) absence of curved chambers; and 4) absence of transverse partitions within chambers (all criteria paraphrased from Antcliffe et al., 2011). We therefore see no need to amend this identification here, especially without a more detailed and focused re-analysis of the material.

Moreover, we note that, as another terminal Ediacaran index fossil (as per the Reviewer), it hardly matters whether the fossils are interpreted as *Shaanxilithes* or *Paleopascichnus* – both would identify this part of the Schwarzrand Subgroup as Ediacaran in age.

- 8) “record not recd”

CHANGES MADE: The text has been fixed as suggested – thank you.

Reviewer 2

- 1) “My only criticism in this regard is the use of the Ecosystem Engineering Impact index of Herringshaw et al. (2017). This index encompasses several metrics, but without actually articulating the different elements in a consistent fashion. For example, Herringshaw et al. (2017) adopted the functional groups of Solan and co-workers without any adjustments. This is problematic because adjustments are needed for applying this scheme to the fossil record when dealing with tiering structure to avoid conceptual overlap. This was discussed by Minter et al. (2016, p. 22), who stated “We omit the categories of epifaunal bioturbators and surficial modifiers (Solan and Wigham 2005) because of their spatial constraints, being covered potentially by any of the other four means of sediment reworking in the epifaunal and semi-infaunal tier”. This overlap compromises the evaluation of the intensity of ecosystem engineering represented by each ichnotaxon in the Ecosystem Engineering Impact approach and as a result the values obtained in this way may be rather arbitrary. My suggestion here is that the authors may

well dispense of the Ecosystem Engineering Impact approach and just use the scheme put forward by Minter and co-workers, which covers the same ground on a more consistent fashion.”

CHANGES MADE: We agree that this is a limitation to Herringshaw et al.’s (2017) EEI method. However, we find that the EEI scheme is useful in addressing the fluid dynamic properties of why some bioturbation behaviours are more ‘impactful’ than others notably from the perspective of bioirrigation potential, particularly as shown by theoretical work by van de Velde and Meysman, (2016). Additionally, we find that it is a useful visual aid for stratigraphically displaying these behaviours, and that it is important to employ a range of methods used by the paleontology, ichnology, and geobiology methods.

We do find it is important to address these caveats and also clarify that the Minter et al. (2017) scheme addresses these issues as well as provides a different framework for analysing ecosystem engineering behaviours. We make this clear with the addition of the following statement to the methods section:

“We note that a limitation to the EEI method is a spatial overlap (e.g., the surficial tier and surficial modifiers) and impossible combinations (e.g., the deep tier and epifaunal locomotion) between tiering and functional group (Minter et al., 2017). However, these limitations were addressed and are accounted for in Minter et al.’s (2017) ecosystem engineering cube scheme.” (lines 176-180)

- 2) “Also, it may be advisable not to include in the analysis those forms which have not been assigned to an ichnotaxon because the behavioural significance of these may be unclear. In any case, looking at the photographs, I would tend to think that those trace fossils illustrated in Fig. 2A may belong in *Helminthoidichnites* the one in Fig. 2B may be referred to as a treptichnid (as the one in Fig. 2E, as indicated by the authors), and the one in Fig. D may be tentatively called *Helminthopsis*. The one in Fig. 2C is simply too fragmentary to say. The ones in Fig. 2F may be plug-shaped burrows as indicated by the authors, but also a preservational variant of a treptichnid (cutting the sample may resolve this issue). The one in Fig. 2G is really unconvincing and I am not even sure if it is a trace fossil. The one assigned to *Planolites* in Fig. 2I is unconvincing, I would place it in *Helminthoidichnites*. The one assigned to *Streptichnus* in Fig. 2K (note that the label is gone) is not convincing either, and it is too partially preserved to suggest an alternative name. In short, some adjustments are needed on this front and some of these will translate into changes in the evaluation of ecosystem engineering.”

CHANGES MADE: We agree with these suggestions. ‘Form A’ and the *Planolites* in Figure 2 have been updated as *Helminthoidichnites*. ‘Form D’ has been updated as *Helminthopsis*. ‘Form B’ and ‘Form E’ have been updated as *Treptichnus*. We have removed ‘Form C’ due to our inability to confidently consistently distinguish it from body fossils. For the plug-shaped burrows which may be in fact be poorly preserved *Treptichnus*, we maintain our initial descriptions on the basis that many plug-shaped burrow slabs cannot be cut as they remain in Namibia (either in outcrop if *in situ* or at the

Ministry of Mines and Energy), and this gives us the most conservative estimate of their ecosystem engineering impact. We have noted this in the following addition:

“Trace fossils broadly interpreted as plug-shaped burrows can be classified as either *Conichnus* or *Bergaueria* when cut open. However, due to the inability to distinguish the two ichnogenera (and, although we did not observe this in samples which were cut and polished, some plug-shaped burrows may be poorly preserved *Treptichnus*; see Jensen et al., 2000) on the bedding plane alone when slabs were not able to be cut open.”
(lines 195-199)

We have also removed the few *Streptichnus* due to the lack of convincing samples, although added the following addition to the discussion noting that *Streptichnus* is found in the Spitzkop even if not noted in this dataset:

“Furthermore, the occurrence of other trace fossils which represent similar complex engineering behaviours which have been reported from the Nama Group but are not included here, such as *Streptichnus narbonnei* in the Spitzkop Member (Jensen and Runnegar, 2005), add to the robustness of these data.” (lines 309-312).

These updated ichnotaxa assignments have been reflected in a rewriting of the trace fossil descriptions in the Electronic Supplemental Material, S1. For these new assignments, we have referenced the Minter et al. (2017) supplemental dataset and Herringshaw et al. (2017) to reassign previously indeterminate forms to their new ichnotaxa assignments and updated the EEI scores and cube occupation spaces accordingly. We also removed the point-counted bioturbation data for ‘Form C’ and *Streptichnus*. This data has been updated in the supplemental material, S5 and S6, and throughout the results, discussion, and results figures. However, because we had largely based our initial EEI and cube assignments for the indeterminate trace fossils on these assignments anyways, the data has not significantly changed enough to change our initial interpretations and conclusions.

- 3) “Finally, the author may wish to cite a recent paper that uses a similar approach to assess ecospace utilization in the interval of the Chapel Island Formation attributed to the *Treptichnus pedum* zone, comparing with the underlying strata in the same area. The reference is:
Laing, B.A., Mángano, M.G., Buatois, L.A., Narbonne, G.M. & Gougeon, R.C. 2019. A protracted Ediacaran–Cambrian transition: an ichnologic ecospace analysis of the Fortunian in Newfoundland, Canada. Geological Magazine.”

CHANGES MADE: Thank you for this reference. We have cited this paper and its results as a comparison made to the observed trends we see in increased ecosystem engineering impact and behaviors:

“This trend in increasing ecosystem engineering prior to the Cambrian is comparable to other Ediacaran-Cambrian sections worldwide, notably in the Ediacaran-aged Blueflower Formation of northwestern Canada where complex burrowing behaviours associated with

deposit feeding are present prior to the Ediacaran-Cambrian boundary (Carbone and Narbonne, 2014) and in the Chapel Island Formation where the trace fossil record preserves a dramatic increase in behavioural innovation and ecospace occupation from the Ediacaran to the early-middle Cambrian (Liang et al., 2019).” (line 315-321).

- 4) “Lines 57-58: some words are missing in this sentence”

CHANGES MADE: Thank-you – we have fixed the text of this sentence.

- 5) “Line 62: for the reader unfamiliar with the recent debate, the first part of the sentence may be hard to understand. I suggest listing the three original hypotheses and then discard the taphonomic one citing the Buatois et al. (2014) paper.”

CHANGES MADE: We agree – we have rephrased this section of text as follows:

“Three major models have been proposed to explain the disappearance of the Ediacara biota: 1) a ‘catastrophe’ model, which proposes a global-scale environmental perturbation analogous to the ‘Big 5’ Phanerozoic mass extinctions (e.g. Amthor et al., 2003); 2) a ‘biotic replacement’ model, which proposes that the extinction was the result of intensifying ecosystem engineering from emerging Cambrian-type metazoan fauna (Laflamme et al., 2013); and, 3) a ‘Cheshire Cat’ model, proposing that the disappearance of the Ediacara biota is instead due to a taphonomic bias from the loss of non-actualistic preservational environments related to the microbial matgrounds (Laflamme et al., 2013). Whereas the ‘Cheshire Cat’ model has been refuted by work showing that Ediacaran-style matgrounds persisted into the Cambrian (Buatois et al., 2014), the ‘catastrophe’ and ‘biotic replacement’ models both possess supporting lines of evidence, as well as critical questions that remain to be addressed (summarized in Tarhan et al., 2018; Darroch et al., 2018a). Taken globally, there appears to be a transition in assemblage composition between the Ediacaran and the Cambrian (Wood et al., 2019), though the influence of potential environmental and taphonomic signals, as well as potential diachroneity in the first and last appearances of particular fossil groups, remains to be constrained.” (lines 62-76).

- 6) “Lines 78-79: I think that the authors meant to cite Mángano & Buatois (2014) instead of 2015 here.”

CHANGES MADE: Thank-you – we have fixed this reference.

- 7) “Line 108: “Spitskop” x “Spitzkop” (and elsewhere in the text and in Figure 1)

CHANGES MADE: We have used the stratigraphic names from Miller’s (2008) Geology of Namibia. Inconsistencies in naming this Member have all been changed to Spitzkop (similarly, “Niederhagen” has been replaced with “Neiderhagen” in the text and figures).

- 8) “Line 124: “trace fossils” x “traces””

CHANGES MADE: Thank you – this edit has been made.

- 9) “Line 158: Using Carbone and Narbonne (2014) here may be complicated because trace fossils in that study are mostly classified in open nomenclature rather than simply assigned to specific ichnotaxa.”

CHANGES MADE: We agree – this reference was primarily used to understanding the bioturbation and feeding behaviours represented by each trace fossil. This was described in lines 180-183: “Functional group, sediment interaction, and sediment modification assignments were given based on supplementary material from Minter et al. (2017) and other literature describing the behaviour of Ediacaran and Early Cambrian trace fossils (e.g., Carbone and Narbonne, 2014).” We have removed the reference here from Line 158 which cited Carbone and Narbonne (2014) as a method of determining ichnotaxa.

- 10) “Line 177: “Trace fossil assemblages present in slabs (n=16) from the Zaris Subbasin were of comparatively low diversity....” x “Trace fossil slabs (n=16) from the Zaris Subbasin were comparatively low diversity.....””

and

- 11) “In contrast, trace fossil assemblages present in slabs (n=61) from the Witputs Subbasin were of relatively high diversity....” x “In contrast, trace fossil slabs (n=61) from the Witputs Subbasin were relatively high diversity....”

CHANGES MADE: These phrases have been updated with the new results from the ichnotaxa and edited for clarity

“Trace fossil slabs (n=13) from the Zaris Subbasin were collected from the Niederhagen, Vingerbreek, and Urusis members. Only two ichnotaxa occur on trace fossil slabs: plug-shaped burrows and *Helminthoidichnites* (electronic supplementary material, table S2; Figure 3a). Trace fossil slabs (n=50) from the Witputs Subbasin were collected from the Kliphoek, Nasep, Spitzkop, and Nomtsas Members. All six ichnotaxa occur in the Witputs Subbasin (occurrences are detailed in electronic supplementary material, table S2; Figure 3b).” (line 202-207)

- 12) “Line 276: “cf.” x “c.f.””

CHANGES MADE: This text has been fixed – thank you.

- 13) “Lines 291-293: It would be useful to back up this statement with some references or more precise facies information.”

CHANGES MADE: We were not clear on this point, which was intended to explain that some environmental/taphonomic conditions may cause an increase in trace fossil diversity. However, a facies analysis specifically focusing on trace fossil preservation and ichnodiversity is not currently well explored in the Nama Group and is out of the scope for this paper. We have made the following adjustment:

“This observed increase in EEIs is a result of the occurrence of treptichnids prior to the appearance of *T. pedum*, while the increase in the number of ecosystem engineering behaviours reflects the increase in ichnodiversity of each stratigraphic member (although, we note that it is possible that this reflects an environmental signal, while the high diversity trace fossil assemblages may instead reflect some change in taphonomic conditions).” (line 304-309)

- 14) “Line 335: This point is also made in Buatois et al. (2018; Scientific Reports), so this paper should be cited here”

CHANGES MADE: Thank-you – we have added this reference.

- 15) “Lines 353-354: The succession studied in Gehling et al. (2001) is actually relatively shallow (see, for example, some of the classic papers by Myrow).”

CHANGES MADE: Thank you; we have revised this statement as follows:

“However, we note that the most diverse trace fossil assemblages, discovered at the base of the Spitzkop Member (including *Parapsammichnites*, Treptichnid-like traces, and a variety of different horizontal burrows), are preserved in thin interbedded sandstones and limestones possessing no evidence for high energy and transport. In addition, trace fossil studies from other late Ediacaran localities have reported Ediacaran-aged *Treptichnus* from both deep (e.g., Carbone and Narbonne, 2014) and shallow (e.g., Gehling et al., 2001) marine environments.” (lines 374-380)

- 16) “Lines 389-391: I strongly suggest toning down this last part of the last sentence. Cambrian levels of ecosystem engineering (particularly since Stage 2) are much higher than terminal Ediacaran levels. There is a fair amount of studies published recently assessing, for example, the increase of bioturbation intensity since Cambrian Age onwards; see Mángano and Buatois, 2014, Proc B; Gougeon et al., 2018; Nature Communications).”

CHANGES MADE: We agree, and would rather like to emphasize that the ecosystem engineering levels in the Nasep and Spitzkop are as high as they are in the very early Cambrian (e.g. reported here in the Nomtsas). We have made the following changes to clarify this:

“Overall, the trace fossil record of the Nama Group records an early increase in ecosystem engineering well before the Ediacaran-Cambrian boundary, which is consistent with the first prediction of the biotic replacement model.” (lines 321-323)

and

“Despite this, our study provides a link between ecosystem engineering and the extinction of the Ediacara biota, provides insights into the evolution of bioturbation prior to the Cambrian, and illustrates that early metazoans were capable of achieving levels of ecosystem engineering approaching those that appear in the earliest Cambrian by the latest Ediacaran.” (lines 415-419)

- 17) “The organizations of the reference list is not consistent with the style of citation through the text.”

CHANGES MADE: Thank you. We have updated the reference list to fix some of the missing and out-of-order references.